# How Can Phenomenology Address Classic Objections to Liturgy?

**Barnabas Aspray**

Pembroke College, University of Oxford, Oxford OX1 1DW, UK; barnabas.aspray@pmb.ox.ac.uk

**Abstract:** Liturgical worship has at times been controversial within parts of the Christian tradition. This article uses phenomenology—especially the thought of Paul Ricœur, Maurice Merleau-Ponty, and Gabriel Marcel—to analyse, evaluate, and respond to five common objections to liturgy by those who reject it: (1) the absence of freedom and spontaneity, (2) the absence of authenticity, (3) the use of symbols to mediate the divine, (4) the use of the liturgical calendar, and (5) liturgy's repetitive nature. This article concludes that those who practice liturgy have something to learn from each objection, but that none of the objections invalidates liturgy. On the contrary, what phenomenology teaches us about the human condition suggests that liturgy is more suitable than forms of worship that try to do without it.

**Keywords:** liturgy; phenomenology; Paul Ricœur; Gabriel Marcel; Maurice Merleau-Ponty; authenticity; spontaneity; symbolism; liturgical calendar; repetition

## 1. Introduction

Ever since the Reformation, Protestants have been divided over whether worship should be liturgical or not, a conflict that continues to this day and has recently been given the name of "worship wars" (Ruth and Lim 2017, p. 20). Yet, the two sides of the debate are not equally represented in academic discourse. While converts to liturgical worship produce numerous arguments in its favour (Galli 2008; Webber 2008; Hunter 2011; Smith 2013; Holtzen and Hill 2016; Wilson 2019; Oxenreider 2020; Bevins 2020), converts to non-liturgical worship are less vocal about their reasons. They tend to be better observed statistically by the explosive growth of Charismatic, Pentecostal, and Evangelical Christianity around the world, of which the vast majority is non-liturgical (Stoll 1990; Anderson 2013; Martin 1994; Stolz and Favre 2019; Moberg and Skjoldli 2018; Hoon 2013).

This article takes the anti-liturgical position seriously on its own terms in order to see what those in favour of liturgy may learn from it, something that has not been done in scholarship so far. Even those who have never encountered these debates and whose traditions take liturgy for granted can be enriched by paying attention to arguments against it. Phenomenology is ideally suited to mediate between opposing views on liturgy, because although these arise from theological convictions about what constitutes worship "in spirit and truth" (John 4:24 NRSV), they necessarily entail beliefs about the constitution of human beings as worshipping creatures. In other words, every theological argument for or against liturgy must presuppose an idea of what it means to be human (i.e., to have understanding, emotions, imagination, habits, and bodies) and phenomenology has a lot to say about what it means to be human, through its focus on the fundamental structures of human experience. Nonetheless, as we shall see, there will be points when phenomenology can go no further and must leave the mediating task to theology.

This article focuses on five classic objections to liturgy, using phenomenology and some theology to address them. It does not pretend to be impartial—it is written by someone who is himself a convert to liturgy—but it does make every effort to treat the opposing arguments fairly and do justice to them. Nor does it pretend to be comprehensive—much more could be said on each of these points. My goal is to provide a sketch of each objection

and a brief insight into how phenomenology reframes it. First, I shall argue that the *order* and *structure* of liturgy is part of our created human constitution, not something to fight against in the name of freedom and spontaneity; yet freedom and spontaneity have their place in a higher order. Secondly, phenomenology helps us to see that the Charismatic movement's contemporary drive for *authenticity* in worship can backfire by forcing worshippers to declare feelings they do not always have. True authenticity is better defined as a correspondence between words and actions, even when feelings are absent. Thirdly, some Protestants, following Calvin, object to liturgy's use of spatial and temporal *symbols* as aids to worship, calling them "crutches of the soul." I will argue that these symbols are necessary and unavoidable, but that this is due to our finitude, not our sinfulness. Fourthly, the liturgical calendar holds the linear and circular aspects of time in tension, and its use of "holy" days is meant to remind us of the holiness of all things. Fifthly, the central feature of liturgy is its *repetitive* nature, which is offensive to those who insist that worship must involve the conscious mind. I shall argue that repetition is not incompatible with conscious engagement, and only loses consciousness when consciousness succumbs to the temptation of allowing itself to be lost. The goal of repetition is to form habits in worshippers that transform their lives, but this cannot be done without the active participation of consciousness. I will conclude by showing that all these things point to the ultimate purpose of liturgy: the displacement of the self and reorientation towards God.

## 2. Freedom, Spontaneity, and Order

I have divided the objections to liturgy into two broad categories, which I have called 'intellectualist' and 'expressivist' based on the alternative vision of worship they propound. Intellectualist worship is focused on doctrinal instruction and the mind, while expressivist worship is focused on feelings and the chance to express a heartfelt connection to God freely. To begin with, I shall address the expressivist objections to liturgy.

Many Protestant Christians object to liturgy on the basis that it does not allow worshippers to be free and spontaneous in their worship. These people see liturgy as a merely human attempt to impose order on worship and to give it structure, restricting the Holy Spirit who should be free to move as he will; in other words, it is an attempt to contain God, but God cannot be contained.

A good example of this objection comes from the Society of Friends, or Quakers, a group whose worship takes non-liturgical forms to their furthest extreme. Spatially, their meetings have "no altar, pulpit, lectern, font or organ". Temporally, there is "no liturgy or programmed order of service with hymns, set prayers and readings; so no prayer or hymn books are required". Their goal in abolishing all these structural elements is so that true "spontaneity" of worship can take place (Gorman 1973, pp. 9, 13). Davies writes that Quaker worship is "anti-liturgical, anti-ceremonial, and anti-sacramental", being instead "an emphasis on spontaneity and inwardness, a waiting of the gathered souls for God to speak to them through the Holy Spirit" (Davies 2015, p. 122). A Quaker would break the silence of a meeting to speak "only when powerfully prompted to do so by the spirit of God" (Gorman 1973, p. 102). For the Quakers "it is manifestly presumptuous of man to attempt to channel that Spirit by set Sacraments, by set times of meeting, by set places of meeting, or by set persons (ordained clergymen). It is not only presumptuous but also futile, as futile as to attempt to catch the wind in a net" (Davies 2015, p. 119). This connotes an anthropology according to which *human order* is incompatible with *divine freedom*. It contains the implicit assumption that the Holy Spirit could never inspire someone to write or perform a liturgy. The Holy Spirit speaks only through spontaneity, and we can never fix down, regulate, or predict what the Holy Spirit will move people to do.

Another form of this objection comes from members of the contemporary Charismatic movement. Wilson describes the typical Charismatic reaction to liturgy as follows:

> To us, the very word liturgy smells of death. It evokes arcane language, disengaged chanting, and dust clouds billowing out of the organ loft. Our version of

> Christianity is about freedom and spontaneity, not empty repetition. We let the Spirit blow where he will, making each meeting different, rather than following the same form of words, every week. We swapped liturgy for liberty a long time ago and have no plans to go back. (Wilson 2019, p. 75)

Part of the answer to this criticism is to note that it is somewhat self-contradictory. If the Holy Spirit is free to move people to do anything, then "anything" must also include liturgy, perhaps even the same liturgy for thousands of years, which is precisely what this view rules out. In its attempt to free the Holy Spirit of all restrictions, this view imposes restrictions on what the Holy Spirit can do. It predicts that God will remain unpredictable.

However, this self-contradiction is only a symptom of a deeper issue which phenomenology can help us to identify. While a phenomenologist such as Gabriel Marcel[1] has some affinity with this criticism of liturgy in his rejection of order and praise of spontaneity, his investigations in dialogue with Ricœur show how we all naturally gravitate towards a certain order and structure over the course of time.

Marcel began his career in violent reaction against the overly ordered systems of thought that he saw in nineteenth-century philosophers. "System means closed totality and complete immanence" (Marcel 1964, p. 250). For him, systematic order meant the death of philosophy because philosophy by its nature is meant to be open, ready to receive whatever new thing comes its way. closure prevents the possibility of any religious revelation from outside. "Suppose that an absolute addition, an entirely unsought gift has been made to man—whether to some men or to all in the course of history," he imagines. The systematic philosopher is compelled to ignore it for fear that it will break apart her system, must "refuse to allow an intrusion to take place in a system regarded as closed" (Marcel [1935] 1949, p. 133). Instead, he writes that "my researches have bearing on all the conditions that permit us to maintain thought in the state of 'openness', in contradistinction to a systematised dogmatics closed in on itself" (Marcel 1952, p. XIII). Marcel's own writing style reflected this pure spontaneity of thought: his first two books are simply publications of the ideas written in his journal, the only order being chronological according to the date on which he had that idea (Marcel [1927] 1952, [1935] 1949).

However, as his writings progressed, Marcel was compelled to admit that a certain order to thinking is both natural and necessary. This happened through an interaction with his close friend and mentee, the phenomenologist Paul Ricœur. When Ricœur wrote a commentary on Marcel's work, he gently chided Marcel for some lacunae in his philosophy which Marcel realised were due to his rather extreme attempts to avoid ordered systems. Marcel wrote to Ricœur admitting that the "missing pieces" were due to his non-systematic approach:

> It is completely true that there are certain problems I have never directly addressed, primarily because I never dreamed of providing a system of philosophy. These are nonetheless points on which I should express myself. (Marcel 1948)[2]

Ricœur later reflected on this episode and pointed to the major difference between them: "As for the systematic spirit Gabriel Marcel cautioned me about, I continue to claim it . . . . I confess I have always needed order and, if I reject any form of totalizing system, I am not opposed to a certain systematicity" (Ricœur 1998, p. 25).

Ricœur equation of systematicity with the "need for order" implies a certain level of disorder in Marcel's philosophy. However, as Ricœur was well aware, Marcel was too great a thinker for his philosophy to be entirely disordered. In a 1968 interview with Marcel, Ricœur acknowledged that "there is no Marcellian system", but instead referred to the "living unity governing all the themes of your philosophy", a characterisation Marcel found more acceptable (Marcel [1968] 1973, p. 251).

---

1  Marcel is not always considered a phenomenologist and claims no strict adherence to Husserl's system. But scholars recognise that Marcel's approach has much affinity with phenomenology (see Gallagher 1962, p. 149; Blundell 2003, p. 100).
2  The Fonds Gabriel Marcel, where these letters from Gabriel Marcel to Paul Ricœur can be found, is at the Bibliothèque Nationale de France, Paris.

Ricœur learnt about the human need for order from the other greatest influence on his early life, the phenomenologist Karl Jaspers. While Jaspers is no fan of closed and totalizing systems, he is aware that for human beings "thinking is by nature systematic. When I think I do not stick to one rudiment of a thought, nor do I merely put thoughts side by side; I relate them to each other" (Jaspers [1932] 1969, vol. 1, p. 276). To try and reject all order is to act contrary to our nature as human beings. Like Ricœur, we all "need order".

What Ricœur, Jaspers, and Marcel show us about the ordered nature of thinking can be applied to other areas of life. The insight that order is an inescapable part of the human condition is confirmed by the history of anti-liturgical movements, which have all acquired a certain order and structure over time. Recall above how Davies said that the Quakers think it presumptuous to "channel" the Spirit by "set times and places of meeting". I doubt that any Quaker community has ever been so extreme as to refuse to plan where and when to meet, trusting that they will all be moved by the Spirit to go to the same place at the same time. However, even if Quakers have done so in the past, they do so no longer. Gorman admits that the Quakers agree to meet at a certain place and time, "usually" on Sunday morning, although he does not comment on whether this agreement is inspired by the Holy Spirit or not (Gorman 1973, pp. 7–9).

Moreover, on inspection, it turns out that the Quakers have a large number of expectations about what the Holy Spirit will and will not move people to do in their worship services. Gorman appears to recognise the contradiction when he says that "the essential spontaneity of Quaker worship . . . makes it difficult to avoid sounding authoritarian when attempting to indicate the kind of ideas and words that are suitable" (Gorman 1973, p. 106). However, this "essential spontaneity" does not prevent him from devoting an entire chapter to describing what is and is not "suitable." We should not be surprised to find that feelings play a role in this judgment. He writes that "the test of a good contribution is the feelings it evokes in the audience: 'Will my contribution help forward the sense of awe, wonder, adoration, praise and thanksgiving—an affirmation of the splendour and goodness of life and its ultimate purpose in life and truth?'" (Gorman 1973, p. 107). Even more tellingly, the opposition between "tradition" and "Spirit inspired" falls apart when Gorman strongly prohibits more than one contribution per worshipper: "it is a well-established custom that any individual speaker only makes one contribution to a particular meeting. It is almost impossible to conceive a situation when departure from this time-honoured tradition would be justified" (Gorman 1973, p. 108). More examples could be given, and Gorman is not unusual in giving many precise restrictions on what should and should not happen in a Quaker meeting. As Davies notes, already in the eighteenth century "the most anti-traditionalist group had almost become entombed in its own traditions" (Davies 2015, p. 121).

Charismatic worship also turns out to have a lot of expectations and restrictions about what is "proper". "In some non-denominational free-church congregations," writes Wolterstorff, "there may be no trace of a liturgical text, nothing written down. Nonetheless, there is a script for doing it right" (Wolterstorff 2015, p. 16). Wilson makes the same point a different way. Although Charismatics may think that "we swapped liturgy for liberty a long time ago and have no plans to go back," he writes, the answer is simple:

> No, we didn't. We didn't get rid of our liturgy; we changed it. . . . If I know that somebody attends a contemporary evangelical or charismatic church, I can make a pretty good guess as to what their order of worship will be, even if I have never been there. (Wilson 2019, pp. 75–76)

Every church that begins by breaking free of liturgy in the name of freedom and spontaneity will gradually form its own routines and structures over time, establishing a culture of what is acceptable in worship. Order—that is, liturgy—cannot be avoided: when pushed out the front door, it creeps in the back door.

However, someone might reply: if people always fall short of the ideal, is that the fault of the ideal, or the fault of our human failings? Should we not lament our tendency to gravitate towards routines? Should we not always strive for the true spontaneity and

freedom of the Spirit? With this question we arrive at the boundary between phenomenology and theology. Phenomenology can only observe "what is" in the human condition, not pass judgment on "what ought to be." With the help of phenomenology, we have observed that "we are liturgical animals", as phenomenologist James K.A. Smith puts it (Smith 2013, p. 12). Human beings cannot help creating routine and order, culture and expectations about what is fitting. Yet, to say that "this is how we are" is not, for Christians, to imply that "this is how we ought to be." Christians believe that human beings are not only created a certain way, but that our created nature has been corrupted by sin. Distinguishing between which aspects of our condition are good and which aspects are corrupted is out of scope for phenomenology. We are led to ask: is our drive for order part of our finite created nature and the way we were meant to be, or is it a corruption of our nature which is meant to be purely free and spontaneous? In other words, is order part of our created finitude or of our sinfulness? This is a theological question and to answer it we must turn to theology.

One helpful way of addressing this question from a biblical or theological perspective comes from 1 Corinthians 14, a favourite among Charismatic Christians because it treats prophecy and speaking in tongues—which are necessarily spontaneous—as normal phenomena of church gatherings. Yet, this chapter concludes with the injunction that "all things should be done decently and in order" (1 Corinthians 14:40 NRSV).

This passage thus shows us two things. First, it shows that for Paul, order is not a sinful drive to be fought against, but part of the ideal to strive for. It is disorder that falls short of the way we were meant to worship. In other words, our liturgical nature arises out of our created finitude, not our sinfulness. We need not be ashamed of our tendency towards structure and setting cultural boundaries on freedom. They are not a blemish to be purged by ever greater efforts; they are a gift from the Creator to be welcomed and used for its rightful purpose.

Secondly, for Paul liturgical order is not incompatible with freedom. Even as he gives instructions about spontaneous contributions, he concludes by saying that order is a necessary part of worship. This challenges the dichotomy between liturgy and spontaneity and suggests that there is a way of holding the two together, meaning that we do not have to choose between them. It "can be attested both by experience and by Scripture," agrees Smail, "that freedom and liturgy are complementary to each other, rather than mutually exclusive alternatives" (Smail 1995, p. 114).

To see liturgy and spontaneity as compatible makes possible a reconciliation of expressive and liturgical worship to the satisfaction of all concerned. If the two are not mutually contradictory, and if both have their place in our created human constitution, then there is nothing to prevent a traditional liturgy from setting aside a period of time for spontaneous contributions. Indeed, that is precisely what we find in the Catholic Charismatic movement that has been growing rapidly for the last fifty years (Csordas 2001; Gooren 2012; Alva 2015; Chappell 2017). Without removing any element of the traditional Roman rite used by Catholics worldwide, Charismatic Catholicism creates space for the elements of freedom and spontaneity typical of the Charismatic movement. By combining order and freedom in this way, the movement creates a church environment in which both lovers of liturgy and lovers of spontaneity may feel at home and yet also be challenged to grow in an area with which they may be less familiar. The Catholic Charismatic movement is an example to all Christians of how both liturgical and expressive worship may flourish in the same church, to the benefit of all.

### 3. Authenticity, Sincerity, and Feelings

The other charge levelled against liturgy by those who favour expressive worship is that liturgy is inauthentic. It does not allow worshippers to express their true feelings about God, either because the language is not their own, or because the liturgy simply ignores feelings altogether and focuses rather on abstract doctrines and formulas. Ruth and Lim describe the Charismatic opposition between liturgy and authenticity thus:

The common characteristic of baby boomers was a questioning of tradition. Just because something is old does not make it right. Just because something has been inherited from the previous generation does not give it value. And just because something has been promoted by a human institution does not make it legitimate. These perceptions easily led to a certain kind of liturgical iconoclasm through which traditional liturgies became suspect in a search for new forms of worship that seemed more authentic. Indeed, authenticity as determined by the worshippers became an underlying ethos throughout contemporary worship. (Ruth and Lim 2017, p. 25)

For many Christians, to recite a "set form" of words in a prayer "deprive[s] men of the capacity of simple prayer to their Creator" (Davies 2015, p. 28). When you pray using someone else's words, you might be articulating the sounds without consciously *meaning* what you are saying. To put this phenomenologically, we have all experienced times when we have repeated someone else's words—e.g., reading a book out loud to someone—without "taking in" what we were saying. This objection to "prewritten prayers" has become prevalent in the majority of Evangelical churches worldwide (Ruth and Lim 2017, p. 102).

Similarly, for the Quakers, worship takes place not when a prewritten prayer is recited, but when the Holy Spirit inspires people to speak by stirring them in their deepest feelings; "the 'secret' inner working of the Spirit is alone acceptable because entirely obedient worship" (Davies 2015, p. 118). As one early Quaker put it: "we wait on the Lord, either to feel him in words, or in silence of spirit without words, as he pleaseth" (Gorman 1973, p. 101). Or again: "when words in meeting [*sic*] reflect what a person feels in the depth of his being . . . then they are likely to 'speak to the condition' of those who hear them" (Gorman 1973, p. 13).[3]

Even those who love liturgy will admit that it can have a tendency to fail to engage the minds and hearts of its participants. Scholars have observed that many have a "tragic feeling of disconnection" between liturgy and the rest of their lives (Rentel 2015, p. 223). Liturgy can be merely "outward show" without any "inner conviction." People can show up at a service, perform the acts, say the words, and go home, while their minds and hearts are elsewhere the whole time. Or indeed, people can attend a liturgical service on Sunday, and the rest of the week go on living lives untouched by the gospel.

A partial response would be to point out that this kind of inauthenticity is equally likely to be found in any other kind of worship. We can never be sure whether the people around us are worshipping sincerely or not, because we can never be sure what they are feeling in the depths of their heart. Each individual can only ever feel his or her own feelings, as phenomenologist Maurice Merleau-Ponty teaches us: "other human beings are never pure spirit for me: I only know them through their glances, their gestures, their speech—in other words, through their bodies" (Merleau-Ponty [1948] 2004, p. 62). We guess the feelings of others by external signs, such as raised voice, shining eyes, open hands, trembling lips, etc. A worshipping community that judges a person's devotion to God by how ardently they feel can only do so based on these signs, and these can always be falsified with sufficient motivation. Consider the example of confession of sin. To be sure, someone can recite a pre-written confession without any inner contrition. However, someone can also give a passionate show of penitence without any inner contrition. Tears and weeping are no guarantee that a person is not acting; they can also be "put on" for the benefit of the observer.

However, this answer misses the primary point of the expressivist objection to liturgy, which is that liturgy invites inauthenticity by giving no place for the worshipper to express his or her feelings. To answer this objection, we must ask whether authenticity is best

---

[3] These two quotations reveal some difference of opinion exists among Quakers as to whether their utterances are directly inspired by the Holy Spirit, or merely the expressed deepest feelings of the human utterer (Gorman 1973, p. 104). We shall discuss this change later. But either way, it is feelings that are being expressed, whether Spirit-inspired or not.

understood as expressing one's feelings, or as consistency between words and actions. Two thought experiments may help us. The first comes from Gabriel Marcel, who records a dilemma in his journal: "I promised C the other day that I would come back to the nursing home where he has been dying for weeks, and see him again. This promise seemed to me, when I made it, to spring from the inmost depths of my being." Yet, later on, he no longer feels like going: "several days have gone by since my visit. . . . And yet I must in honesty admit that the pity I felt the other day, is today no more than a theoretical pity." His deepest feelings no longer move him to make a visit; if he went ahead, it would be out of a cold recognition that he ought to keep his promise, not out of a genuine feeling for his friend: "I have to recognise that this impulse no longer exists, and it is no longer in my power to do more than imitate it by a pretence which some part of me refuses to swallow" (Marcel [1935] 1949, pp. 48–49). However, what exactly did he promise? To feel the same way in a few days as when he made the promise? That would be foolish, as he cannot possibly control how he will feel in the future. What he committed to was an *action* which faithfulness requires him to perform regardless of his feelings about it. Oddly enough, it turns out that to be faithful to his friend requires him to be *in*authentic, if authenticity is defined as being true to one's feelings.

The second thought experiment concerns two married men. The first tells his wife "I love you," and his words correspond truly to his feelings of the moment. He is overcome with passionate desire for her. However, his character is weak. A few days later he is overcome by desire for another woman and commits adultery with her. The second man tells his wife "I love you," even though he does not feel it at that moment—in fact, he is feeling a growing desire for another woman. However, however strongly his feelings urge him, he is never unfaithful to his wife. Which man's words were the more authentic?

These two thought experiments show that there are two kinds of authenticity: being true to one's feelings or being faithful to one's promises. Following Milnes and Sinanan (2010), I suggest that we call the first *sincerity* and reserve authenticity for the second.[4] While authenticity has always been praiseworthy throughout history and in all cultures, sincerity was not counted a virtue in the West until modernity (Trilling 1972). Moreover, as the above two thought-experiments show, sincerity and authenticity can sometimes be opposed, so that one must choose between them. As Merleau-Ponty notes: "if sincerity is one's highest value, one will never become fully committed to anything, not to a Church or to a party, not to a love or a friendship, not even to a particular task; for commitment always assumes that one's affirmation surpasses one's knowledge, that one believes by hearsay, that one gives up the rule of sincerity for that of responsibility" (Merleau-Ponty 1964, p. 179). In other words, if we are only ever faithful to our feelings, then we cannot be faithful to anything else, because our feelings are liable to change all the time.

Based on these phenomenological insights, I contend that worship is inauthentic when a person's words fail to correspond, not to his or her feelings, but to his or her lifestyle and daily choices.[5] This means that inauthenticity is equally possible for both kinds of worship, expressive and liturgical.

What about the charge of insincerity, the real expressivist objection to liturgy? If this is an accusation that worshippers may falsify their feelings for outward show, then an interesting reversal takes place. Liturgical worship cannot be sincere or insincere, because it does not typically invite the worshipper to express his or her feelings at all; the words of liturgy are usually about what is held true by the worshipping community regardless of how they happen to feel about it. However, with expressive worship, insincerity becomes

---

[4]　This distinction is not the same as the one made by Wolterstorff (2018, pp. 117–20); but apart from terminology, Wolterstorff comes to similar substantive conclusions to the ones in this article.

[5]　Granted we all make mistakes, and none of us perfectly aligns our words with our lifestyle. That is perhaps why Nathaniel Marx suggests that authentic liturgy is "minds in tune with voices" (Marx 2020), placing the emphasis on our will rather than our actions. Yet a weak-willed person might mentally assent to the words of liturgy while speaking them and yet still fail to practice its precepts in daily life. My position assumes that we have some control over our actions, and that if we truly desire to live a certain way, we will make some progress towards such a lifestyle, however small.

not only possible but likely, since expressive worship encourages worshippers to express feelings they may or may not have. In expressive worship I am compelled to sing about my joy, love, and gratitude even if I am feeling none of these things. As Smail puts it, "charismatic worship . . . can often highlight our subjective response to God and neglect the objective work of salvation that Christ undertook on our behalf" (Smail 1995, p. 114). He adds that "worship must have a place not just for the moments when hearts lift high and eyes are shining and joy abounds, but for the dull days when we are empty and unresponsive in ourselves" (Smail 1995, pp. 111–12). This is precisely what liturgy provides. In liturgy, I can confess that God is glorious and that I am sinful, not as emotions, but as realities that do not depend on me, and the truth of my confession will be tested by my way of life.

This leads us to a tension that is difficult to resolve to everyone's satisfaction, and may explain the large number of conversions from each side to the other of the aforementioned "worship wars". Some people want to worship in an expressive way, not because it guarantees the absence of hypocrites, but because it enables the individual worshipper to connect with God at an emotional level. For these people, liturgy feels cold and empty compared with the warmth of feeling given by an expressive worship service. Yet, for others, the emotional nature of expressive worship seems false and coercive, and they are relieved to discover in liturgy that God does not require them to feel a certain way. They take comfort in the fact that their faith is founded on an objective reality that does not depend on their feelings, and that their worship is acceptable if they are making every effort to live in a way that pleases God. Smail, for example, writes about how much he appreciated a period of attending a liturgical church, because "I was not under any pressure to shine with joy or glow with gifts, but was constantly reminded in the sacrament that, however I might be feeling or faring, what Christ had done on Calvary was done for ever and was available for me" (Smail 1995, p. 112). Another anonymous theologian once recounted to me that as a child he attended expressive worship services, where he felt under pressure to be joyful, exuberant, enthusiastic, and grateful to God every Sunday. When as a young adult he went through a difficult time, he did not find such feelings welling up out of him and began to experience an alienation from the style of worship that demanded this. One day he walked into a liturgical service, where "I was told to bow my head because God was holy, and it had nothing to do with how I felt about him." This was his first experience of a style of worship that prioritised the objective reality of God's holiness regardless of how he felt about it, and it converted him on the spot.

How can liturgy satisfy the desire of some people to express their emotions in worship, and at the same time remain focused on objective reality that does not depend on our emotions? I suggest that the answer lies in seeing emotion, not as an uncontrolled reaction in the moment, but as something we can direct over time. We must correct the tendency of some liturgical texts which "give the impression that emotions or feelings are a bad thing, to be suppressed or eradicated" (Gschwandtner 2019, p. 126). Liturgical services can seem dry, and it can be difficult for those who love the vibrancy of expressive worship to adjust to them. However, if they are willing to put the effort in to adjust, they will find that liturgy aims, not to suppress emotion, but to channel and train it. This is well illustrated in a story about some Jewish congregants who complained to Rabbi Abraham Heschel that "the liturgy did not express what they felt. Would he please change it? Heschel wisely told them that it was not for the liturgy to express what they felt, it was for them to learn to feel what the liturgy expressed" (Byars 2008, pp. xvi–xvii). Liturgy does not leave us to express whatever emotions we currently have, as if the only purpose of worship was self-expression like a sort of group therapy. Liturgy challenges us to develop our emotional lives and be transformed into the likeness of Christ; it "helps us to express and redirect our emotion in healthier ways and to train our pathos in a new manner" (Gschwandtner 2019, p. 134). Its goal is to reorient every part of our lives, including our emotions, so that we reflect the fullness of mature humanity. No doubt this takes effort, and people are often put off by the demand made on them to channel their emotions rather than freely express them.

However, as we shall see, the ultimate goal of liturgy is to take our focus off ourselves and put the focus on God, the objective reality of who he is. When we have become so constituted that God is the centre of our lives, we will find that our emotions follow suit; they are welcome passengers, but no longer in the driving seat.

### 4. Are Liturgical Symbols "Crutches of the Soul"?

Thus far, we have looked at objections to liturgy that come from those who favour styles of worship that are free, authentic, and spontaneous, like the Charismatic movement and the Quakers. We now turn to a different kind of objection, coming from a form of worship I call intellectualist. This form of worship has the goal of beholding God with the mind's eye alone through the conveyance of sound doctrine in hymns and preaching. Its objection to liturgy is that liturgical symbols—spatial symbols, like holy objects, icons, or buildings, and temporal symbols, like special days or weeks set aside and consecrated as holy—are obstacles to this pure, unmediated communion with God.

What is wrong with the use of symbols to mediate the divine? According to both Quakers and Calvinists, they are "crutches of the soul" (Davies 2015, p. 118); in other words, a healthy soul can and should encounter God without them. A mediation is by its nature something that stands between us and God, and is thus forever in danger of becoming an idol, diverting our attention from God to itself. We must therefore abolish all mediations, allowing us to encounter God "immediately" without the need of aids.

In Protestantism, the most influential source of this anti-liturgical theology is John Calvin. Calvin was supremely iconoclastic, having rejected the seventh ecumenical council that permitted the veneration of icons. He sought to purge worship of all material props, leaving the worshipping soul naked in the immediacy of encounter with God. To this pure immediacy of encounter there is one exception, one permitted mediator of the divine: the proclamation of the Word of God. While "Word of God" refers properly to Christ, the primary access to Christ is the Bible. Hence, for the Puritans, a group strongly influenced by Calvin, "the chief regular means of grace seems to have been the Sermon" (Davies 2015, p. 31). The priority of the Word is reflected spatially in the architecture of Puritan churches, which give the most prominent position to the pulpit from where the Scriptures are read and the sermon is preached. It is also reflected temporally in the order of service, where the sermon is typically the longest part of the meeting: "what was chiefly important for Puritans and Nonconformists was the faithful reception of the Divine Word in the lengthy Scriptural readings . . . and their equally lengthy expositions" (Davies 2015, p. 32).

While Quakers agree that the mediation of symbols is wrong and unnecessary, they differ from Calvinists in their view of what remains after the symbols have been removed. As we have seen, for Calvinists it is the Scriptures, but for the Quakers, it is the witness of the Holy Spirit: "the 'secret' inner working of the Spirit is alone acceptable because entirely obedient worship" (Davies 2015, p. 118). In other words, the two groups have different answers to the question, "what is the point of contact in our human condition by which we encounter God?" For Calvin, it is the intellect; for the Quakers, it is feelings. Recall the two citations from Quakers above: "we wait on the Lord, either to feel him in words, or in silence of spirit without words, as he pleaseth" (Gorman 1973, p. 101); and "when words in meeting [*sic*] reflect what a person feels in the depth of his being . . . then they are likely to 'speak to the condition' of those who hear them" (Gorman 1973, p. 13).[6]

What does phenomenology say about these drives towards immediate contact with God? To answer this question we turn to the hermeneutic branch of phenomenology in its primary representative, Paul Ricœur. Ricœur's hermeneutics was founded on the insight that all knowledge and experience is mediated through the concepts, categories, and plausibility structures given us by our culture and upbringing, our situatedness in time and space. Whenever we read or hear any communication, we cannot help appropriating it in

---

6 These two quotations reveal some difference of opinion exists among Quakers as to whether their utterances are directly inspired by the Holy Spirit, or merely the expressed deepest feelings of the human utterer (Gorman 1973, p. 104). We shall discuss this change shortly. But either way, it is feelings that are being expressed.

light of our unique life-experiences. When we think we are experiencing something directly, it only means we are not conscious of the mediating lens that filters our experience and interprets it for us. Even before we consider God or the world, Ricœur says that our own self-understanding is "mediated by a universe of signs." Indeed, "all reflection is mediated, there is no immediate self-consciousness" (Ricœur 1977, pp. 27–28). In other words, any idea of immediate communication from God is impossible for the finite human condition (see also the discussion in Westphal 2009, pp. 17–26). As Gschwandtner puts it, "there is no unmediated phenomenological experience of 'revelation'" (Gschwandtner 2019, p. 119).

The history of the Quakers and the Calvinists provides strong evidence in support of this foundational tenet of hermeneutic phenomenology. At the start of the Quaker movement all contributions during worship were considered to be prompted directly by the Holy Spirit (Gorman 1973, p. 101). This was a very bold claim, since it meant that a contribution could not be questioned or criticised: God himself was speaking. However, nowadays Quakers use more cautious language. They no longer claim that a contribution is directly inspired by God, but only that it is an expression of the contributor's own deepest feelings. This makes it possible to criticise a contribution as inappropriate (Gorman 1973, p. 104). Feelings are not an infallible guide to the will of God.

The Calvinist or Reformed tradition has gone on a similar journey with regard to the hermeneutic insistence on mediation. At the time of the Reformation, the Word of God was seen as plain, needing no mediation for those who read it by faith. Since that time, history has witnessed countless church splits and new denominations, each one claiming that it alone is reading Scripture according to its plain meaning in contrast to all the others. This testifies to the many possible interpretations of Scripture, and shows how powerful can be the influence of our cultural lens, especially when we are unaware of it and think we are seeing things simply "as they really are". Like the Quakers, the Reformed tradition has changed over time and is now more modest about its claims. For example, Karl Barth, the greatest Reformed theologian of the twentieth century, insists that everyone reads Scripture through the mediation of their own worldview: "there has never yet been an expositor who has allowed only Scripture alone to speak" (Barth 2010, p. 727).

Thus the use of symbols to mediate the presence of God is the equivalent for intellectualists to the use of order and routine for expressivists. In both cases, phenomenology has shown that these things cannot be avoided; they are built into our human constitution. However, phenomenology cannot determine whether their unavoidability is a defect or something to embrace about ourselves—in other words, whether it is due to our sinfulness or our finitude. To ask this question we must once again go beyond the boundaries of phenomenology and enter the domain of theology.

Most Reformed theologians who recognise the need for mediation and symbols lament this as a weakness in our human condition. When it comes to reading/hearing the Word of God, Barth maintains that unmediated interpretation is the ideal from which we fall short due to our imperfection. He draws a direct comparison between the fact that we are situated by a philosophical worldview and the fact that we are sinful, writing that "it is no more true of anyone that he does not mingle the Gospel with some philosophy, than that here and now he is free from all sin" (Barth 2010, p. 729). For Barth, it is our sinfulness that makes mediation inevitable.

James K.A. Smith takes another view. Smith is not only a phenomenologist but also a convert to liturgy from the Calvinist tradition. *Contra* Barth, Smith argues that it is our finitude, not our sinfulness, that makes mediation unavoidable: "interpretation . . . is an inescapable state of affairs that accompanies the finitude of creaturehood and, since it is an aspect of creation, is 'good'" (Smith 2012, p. 24). If mediation is due to our finitude, making symbols a necessary part of worship, then liturgy must be seen as a simple acknowledgement of our finitude, rather than an acceptance of our sinful state.

In all these areas, then, we find that it is better to see liturgical symbols and mediations as an embrace of our finitude, not a concession to our sinfulness. To call them "crutches of the soul" is to imagine ourselves as more than human, as if the wings of an airplane were

"crutches" because we ought to be able to fly without them. We should not be ashamed that we cannot do without them, but should rather make use of them as means to access the divine.

## 5. The Liturgical Calendar and "Holy Days"

Let us focus on a particular example of a mediating symbol that has drawn heavy criticism: the liturgical calendar, which introduces a cyclical element to time as well as designating some days of the year as "holy days" (which is where the English word "holiday" comes from). Many biblical theologians consider the cyclical view of time to be pagan, in contrast to the linear view of time that was introduced by the Judeo-Christian tradition. According to the twentieth-century biblical theology movement, the religions of the ancient Near East apart from Judaism saw time as endless repetition without change or development: "all evidence points to there having been, in the earliest religious thought, a vision of the cosmos that was profoundly cyclical" (Cahill 1998, p. 5). But "the Jews were the first people to break out of this circle" (Cahill 1998, p. 5). The Hebrew Bible replaced the cyclical view of time and history with a linear one. The world has a beginning and an end, and above all a God who acts in world history, thereby changing it irrevocably. The emphasis on the linearity of history for Christians stands in contrast to the way liturgy emphasises its circularity.

However, there is a still clearer objection. For many Protestants, the elevation of special days is explicitly forbidden by Scripture. The apostle Paul writes to the Galatians: "how can you turn back again to the weak and beggarly elemental spirits? How can you want to be enslaved to them again? *You are observing special days, and months, and seasons, and years.* I am afraid that my work for you may have been wasted" (Galatians 4:9–11 NRSV, italics added). This passage shows how urgently Paul thinks the Galatian church to be making a huge mistake that undermines the Christian gospel. Understood in this way, the liturgical calendar strikes at the heart of the Christian message.

A well-articulated example of this objection comes from A.W. Tozer, a twentieth-century American pastor and household name for Evangelicals. Tozer claims that, in the Hebrew Bible, God taught the Israelites about his own holiness by designating certain objects, days, and places "holy": "There were holy days, holy vessels, holy garments. There were washings, sacrifices, offerings of many kinds. By these means Israel learned that God is holy. It was this that He was teaching them. Not the holiness of things or places, but the holiness of Jehovah was the lesson they must learn" (Tozer 1948, p. 124). However, these designations were a temporary pedagogical tool, to be abandoned when the full revelation came:

> Then came the great day when Christ appeared. Immediately He began to say, "Ye have heard that it was said by them of old time—but I say unto you." The Old Testament schooling was over. When Christ died on the cross the veil of the temple was rent from top to bottom. The Holy of Holies was opened to everyone who would enter in faith. . . .
>
> Shortly after, Paul took up the cry of liberty and declared all meats clean, every day holy, all places sacred and every act acceptable to God. The sacredness of times and places, a half-light necessary to the education of the race, passed away before the full sun of spiritual worship.
>
> The essential spirituality of worship remained the possession of the Church until it was slowly lost with the passing of the years. Then the natural legality of the fallen hearts of men began to introduce the old distinctions. The Church came to observe again days and seasons and times. (Tozer 1948, pp. 124–25)

It is clear that this last sentence is a devastating criticism when we recognise its reference to Galatians 4:10. If the Church is "observing days and seasons and times", this means, for Tozer, that the Church has abandoned the Christian gospel. The liturgical calendar is but one example of human weakness that forever creeps back into "the fallen

hearts of men" (as Tozer puts it). The gospel abolishes all distinctions between days, places, and objects, consecrating everything alike as holy.

On the other side, liturgical scholars defend the liturgical calendar by rejecting the claim that Judeo-Christian time is linear. Andrew Louth writes that it is a "'vulgar error', popular among some theologians, that cyclical time spells meaninglessness, in contrast with the purposeful nature of linear time" (Louth 2013, p. 82). Gschwandtner adds that "in its liturgy, its cycles of fasts and feasts, Christianity (and also Judaism) remained cyclical, presumably precisely because the cyclical was indeed meaningful" (Gschwandtner 2019, p. 33). The linear historical and eschatological tenets of the Christian faith do not "reflect the reality of liturgy, which is lived in its cyclical nature as what concerns us now and celebrates the events (such as those of Christ's life) as mattering now and always anew" (Gschwandtner 2019, p. 54).

In an effort to reconcile these conflicting claims, I contend that the two views of time are not so incompatible they appear, and that both are present in the Judeo-Christian worldview. The biblical evidence shows that Judaism had a liturgical calendar of feasts and fasts, so it cannot be entirely true that the Jews "broke out of the circle" of time, as Cahill suggests (Cahill 1998, p. 5).[7] In fact, Judaism took up the Ancient Near Eastern circularity of time and incorporated it into a broader arc of linear history. We can see this by noting a key difference between Jewish and pagan annual celebrations. The pagans celebrated seasonal events, like the harvest, the new moon, the solstice, the equinox, etc. While Judaism also celebrated some of these, it also had memorials of events that had taken place once and changed the world forever, such as the Exodus out of Egypt (Passover), or the rescue from annihilation by the Persians (Purim). These were signs of God's faithfulness and his ability to change the course of history.

*Contra* the liturgical scholars' claims, Christianity has a linear view of time which it adopted from Judaism. It is an inescapable part of the Christian account of reality that God is redeeming the world in an irrevocable way; one day history will be over and it will be the end of all things: no more evil, no more suffering, no more death, no more tears (Revelation 21:4). Circular time is not meaningful by itself, as Louth suggests. If time were truly circular, as with Nietzsche's "eternal recurrence," then life would be meaningless because nothing anyone did—nothing even God did—would make any permanent difference. It would mean that the evil elements we see in our world today are just as eternal as the good, leading to a dualism that is far from the Christian account of reality. Likewise, Gschwandtner rightly points out that we celebrate these events annually to make them meaningful for today, but we must also note that part of the meaning of the events of Christ's life is that they happened once in history and changed the world forever. They are meaningful as annual celebrations only *because* they really happened once and we are different than we would have been had they never happened.

However, *contra* Tozer, it was not a corruption of Christianity also to adopt the circular aspect of time from Judaism. Christianity began as a branch of Judaism that believed the Messiah had come in the person of Jesus of Nazareth. Most Jews believed that when the Messiah came it would signal a whole new order as significant as the founding of the people of Israel under Moses at Mount Sinai. Therefore, it was not in spite of, but because of their Judaism that the early Christians stopped practising the Jewish liturgical calendar. Within the Jewish way of thinking, to continue to practise these old rituals would have amounted to denying that Jesus was the Messiah. This is the crucial context for understanding Paul's frustration in Galatians 4:9–11. The "special days, weeks, months and years" he was talking about were *Jewish* ones, which everyone would have understood in context. For those Judaizers in Galatia, keeping the Jewish liturgical calendar amounted to denying the lordship of Jesus and rejecting the gospel. Paul's words were never meant to prevent later Christians from creating a new set of *Christian* feasts and fasts to help commemorate the decisive events of the New Testament, i.e., of Jesus' life, death and resurrection.

---

[7] For the annual Jewish feasts and fasts, see Leviticus 23:3–42.

Why is the liturgical calendar so important? Precisely because it meets us in our finitude and does not expect more of us than is possible. It contains a deep phenomenological insight about the human condition: that our brains are limited, and we can only dwell on one or two things at once. Over time, we tend to dwell more and more on a limited set of things and gradually forget about the others. Gabriel Marcel describes the phenomenon in this way:

> As my life becomes more and more an established thing, a certain division tends to be made between what concerns me and what does not concern me, a division which appears rational enough in the making. Each one of us thus becomes the centre of a sort of mental space, arranged in concentric zones of decreasing interest and decreasing adherence . . . . This is something so natural that we forget to give it any thought or any representation at all. (Marcel [1935] 1949, pp. 70–71)

Over the course of our lives, Marcel is saying, we become increasingly interested in a finite set of things and correspondingly uninterested in anything that falls outside that set. This is a natural human tendency, yet in this case it is not something liturgy encourages us to accept as part of our finitude: rather, it is something that liturgy is designed to correct. Without the guidance of a liturgical calendar, we gradually begin to over-emphasise some parts and forget other parts of Christian doctrine. We can become obsessed with one or two fragments of the gospel to the neglect of the rest. Our faith can become ever increasingly austere, fasting but never feasting; or it can become ever increasingly indulgent, feasting but never fasting. The liturgical calendar is a way of balancing our spiritual diet[8] by ensuring that we both feast and fast, lament and celebrate; it helps us remember the Trinity, the Incarnation, the Resurrection, the outpouring of the Holy Spirit, and all the other key Christian doctrines in turn without any effort on our part. The liturgical year is full of ancient wisdom, making sure we do not leave anything out that is important for our spiritual health. That is also why it's a mistake to imagine that observing special days is "works-righteousness" as Tozer believes. The liturgical year was never intended as a way of earning our salvation or of becoming righteous before God by our special efforts of observance. It is for our *sanctification* that the liturgical year exists, to help us grow in well-rounded spirituality. Additionally, all of this presupposes that our unavoidable tendency to celebrate anniversaries is part of our created finitude, not sinfulness.

Likewise, from a liturgical perspective, our tendency to treat some days as special is an aspect of finitude, not sinfulness. It is a strange fact, but phenomenologically observable, that when we abandon the idea of "holy" places, times, and things, we do not start to treat everything as holy—instead we default to treating nothing as holy. As Wolterstorff observes about worship in general, "it has been my experience that those who declare that all of life is worship almost always downplay the importance of what I am calling worship" (Wolterstorff 2015, p. 41). These people misunderstand the subtleties about how language works, both in our lives and in Scripture. They are like the man who stopped calling his wife "my beloved" because it implies he does not love his children, not realising that he loves his children best *by* loving his wife first. Similarly, we need "holy" days precisely to remind us that everything is holy to some degree. That is why Genesis 2:3 records that God "made the Sabbath day holy"—not because the other days are not, but that the special holiness of the Sabbath is a discipline to help us remember the holiness of all things.

## 6. How Repetition forms Virtuous Habits

The liturgical calendar is but one instance of the biggest and most contentious issue with liturgy: its repetitive nature. Those who oppose liturgy are concerned that oft-repeated

---

8    Janco objects to the prevalence of food metaphors in worship studies, on the basis that "they present liturgy as a thing to be consumed, rather than an activity to be performed" (Janco 2009, p. 49). But the food metaphor cannot be so easily dismissed. It is true that we participate in the liturgy, but it is also true that its spiritual sustenance comes from elements we did not create or choose. Similarly, we participate our own food consumption when we cook, yet what is decisive for the health of our meal is the ingredients which we did not create ourselves.

words and gestures will become a performance in which the mind is not active. To repeat the same words again and again can rapidly become automatic, an act which we perform without meaning what we are saying. The repetitive character of liturgy is the reason it has been accused of cultivating hypocrisy, legalism, and (perhaps the most common accusation) of simply being boring, failing to engage the interest of participants.

In what follows I argue that the goal of repetitiveness in liturgy is the formation of virtuous habits that transform the basic disposition of the human person at a preconscious level. This habit-formation is never intended to abolish the use of the conscious mind, but rather to serve consciousness as a tool. Therefore, the charge levelled against liturgy—that it destroys conscious engagement—points not to an inevitable feature of liturgy, but to a perennial temptation that accompanies liturgy, into which liturgical worshippers have often fallen.

According to the phenomenologist James K.A. Smith, the difficulty many people have with liturgy's repetitiveness comes from a modern intellectualist worldview that sees the human person as a "thinking thing":

> [For the intellectualist,] *what* I am is an essentially immaterial mind or consciousness—occasionally and temporarily embodied, but not essentially. This bequeaths to us a dominant and powerful picture of the human persona as fundamentally a thinking thing—a cognitive machine defined, above all, by thought and rational operations. (Smith 2009, p. 42)

Learning and formation are thus simple procedures for the intellectualist: a teacher conveys information to a student, like a computer downloading a new program. When the student knows what is right, he or she will thenceforth act accordingly, or else be condemned as a hypocrite. This model of learning is the basis for intellectualist forms of worship:

> It is just this adoption of a rationalist, cognitivist anthropology that accounts for the shape of so much Protestant worship as a heady affair fixated on 'messages' that disseminate Christian ideas and abstract values (easily summarized on PowerPoint slides). The result is a talking-head version of Christianity that is fixated on doctrines and ideas. (Smith 2009, p. 42)

This is a gravely misguided way of construing the human person, Smith continues, yet it is prevalent in intellectualist models of worship: "because the church buys into a cognitivist anthropology, it adopts a stunted pedagogy that is fixated on the mind. . . . As a result, significant parts of who we are—in particular, our noncognitive ways of being-in-the-world that are more closely tethered to our embodiment or animality–tend to drop off the radar or are treated as nonessential" (Smith 2009, pp. 43, 46). In other words, the church has adopted a truncated picture of what it means to be human, which has had a detrimental effect on our forms of worship. We focus only on the way our minds affect our bodies, and ignore the way our bodies can affect our minds in return. We focus only on knowledge as information transfer, ignoring the effect of the imagination in shaping our desires and inclinations at a far deeper level than simply what facts we know.

Let us illustrate intellectualist pedagogy by a scene from the movie *Those Magnificent Men in their Flying Machines* (Annakin 1965). It is 1910, and a German army officer is commissioned to fly in an international air race from London to Paris. On the day of the race he falls sick, and his captain steps in to replace him. The captain has never flown before, but this does not trouble him because he has the book of instructions. He somehow manages to get the plane off the ground, the manual flapping wildly in the wind, while he flips its pages frantically to find out how to make the plane go higher and faster. However, when he is over the English channel, the wind tears the book out of his hands and he is left helpless, soon afterwards to crash-land in the sea.

The captain's error is obvious: when we say a pilot "knows how to fly," we do not mean that she has memorised the instruction manual. We mean rather that she has trained her hands and eyes to acquire the right disposition and inclination—in other words, *habits*

that enable her to fly well. The only way to acquire these habits is by constant repetition over a long period of time. Without such repetition, she will never acquire the competency of a skilled pilot.

Yet, Smith argues that non-liturgical forms of worship are trying to do just what that German captain did: to imagine that a person can be transformed simply by being given the right information, without taking into account his or her inclinations, dispositions, taste, and tendencies that join mind and body in an inseparable unity. It is to ignore centuries of wisdom contained in Christian liturgical traditions:

> Having fallen prey to the intellectualism of modernity, both Christian worship and Christian pedagogy have underestimated the importance of this body/story nexus—this inextricable link between imagination, narrative, and embodiment— thereby forgetting the ancient Christian sacramental wisdom carried in the historic practices of Christian worship and the embodied legacies of spiritual and monastic disciplines. Failing to appreciate this, we have neglected formational resources that are indigenous to the Christian tradition, as it were; as a result, we have too often pursued flawed models of discipleship and Christian formation that have focused on convincing the intellect rather than recruiting the imagination. (Smith 2013, p. 39)

The phenomenology of habit is the key that can unlock the wisdom implicit in traditional liturgies. It can enable us to develop a fuller anthropology that recognises the role not only of the intellect, but also of the body, the imagination, and history: as Merleau-Ponty writes, "the phenomenon of habit in fact leads us to rework our notion of 'understanding' and our notion of the body" (Merleau-Ponty [1945] 2013, p. 146). Habit reveals the deep mutual influence between mind and matter, in which consciousness may sediment itself into semi-conscious decisions and actions that look almost exactly like instincts—but they are not, because they were given by consciousness and still depend on consciousness for their activation.

Habit is not merely an operation of the mind. It is a mistake to think of a habit as merely a series of conscious calculations made at lightning speed. For example, when a blind man uses a cane to discover the objects around him, he does not first feel the pressure on his hand and then rationally conclude that there is something at the other end of the cane: "habit does not consist in interpreting the pressure of the cane on the hand like signs of certain positions of the cane, and then these positions as signs of an external object," says Merleau-Ponty, "for the habit *relieves* us of this very task" (Merleau-Ponty [1945] 2013, pp. 153–54, italics original). The blind man knows the pressure on the hand as the presence of an object, just as a sighted person knows a few simple lines on a page as the depiction of a face before any conscious thought has taken place.

However, neither is habit merely a bodily process, where "the body" is thought of as separate from "the mind." Rather, the deep interconnection of mind and body is revealed in the very fact that the mind itself can have habits, inclinations, dispositions given to it by forms of knowledge: "knowledge, no less than motor conduct, comes from habit" (Ricœur [1950] 1966, p. 292). Knowledge does not sit in the mind merely as an accumulation of data, piled up and arranged like objects in a warehouse. Some knowledge is held almost unconsciously as the lens by which we perceive the world. Like how to fly an airplane, knowledge is that *through which* we engage with the world:

> What I know intellectually is present to me the same way as the bodily skills I have. What I learn, what is understood in an original act of thought, is constantly being left behind as an act and becomes a sort of body of my thought: thus knowledge becomes integrated with the realm of capabilities which I use without articulating them anew. Each time I form a new thought I call up some old knowledge without being aware of that knowledge as such. We could say that "knowledge" is that which I do not think, but by means of which I think. (Ricœur [1950] 1966, p. 294)

We can become accustomed to thinking a certain way by repetition. Certain thought processes and channels come easily, while less commonly used ways of thinking require effort (Kahneman 2012).

Ricœur's point, that knowledge functions as a disposition through which we see the world, explains why liturgy repeats core Christian doctrines so regularly. Its aim is to implant these doctrines deep into the worshipper, no longer simply as cognitive information, but as a foundation for all other thought and action. It is not enough simply to hear once that Christ is both God and man: that will not make it the keystone of one's life. If we want this knowledge to be "written on our hearts" (cf. Deuteronomy 11:18) then we must say it to ourselves again and again until we indwell it.

If we view liturgy as habit-forming repetition, we are equipped to answer one of the controversial questions among liturgical scholars about the relationship between liturgy and the rest of life. It is evidence in favour of those, like Gschwandtner, Smith, and Ratzinger, who argue that "liturgy trains us to act differently in the world" (Gschwandtner 2019, p. 196). For the early and medieval church, formation in Christlike virtue was the goal of the entire Christian life (Wilken 2005, pp. 268–90), and virtue was simply the name for a good habit (see, e.g., Aquinas 2018; *Summa Theologiae*, I.II 55.1). Liturgy is repetitive because its goal is to cultivate virtues, i.e., good habits, dispositions towards the good, which have a transformative effect on the life and actions of worshippers so that they go out into the world and make a difference. The outward-focused nature of liturgy is most likely why the earliest Christians chose the Greek word *ekklesia* for their liturgical gatherings. Hurtado notes that it was not the most obvious word to use:

> There were a number of frequently used terms available such as *thiasos* (the characteristic term for a group of persons who associated for the worship of a particular deity), *eranos* (a fellowship to hold religious feasts to which participants contributed), *koinon* (a fellowship), or *synodos* (a group following a particular teaching). . . . *Ekklesia* designated the gathering of citizens of a city to conduct civic business. (Hurtado 1999, p. 54)

*Ekklesia*, therefore, indicates an outward-looking orientation, a community gathered not for its own sake, but to transform its members so that they can transform society. Just as the word used today by the Catholic Church, "mass," originates from the word for "mission," so the early church saw its gatherings as oriented towards the outside world. The liturgy is there to refocus and reorient us for the purpose of our lives lived outside the church building. The repetition in liturgy is therefore the result of a deep, ancient wisdom that knew how necessary habit-formation is for our inner transformation. As Ratzinger summarises:

> Ultimately, it is the very life of man, man himself as living righteously, that is the true worship of God, but life only becomes real life when it receives its form from looking toward God. [Liturgy] exists in order to communicate this vision and to give life in such a way that glory is given to God. (Ratzinger [2000] 2014, p. 18)

With the nature and purpose of habit in mind, let us reconsider the charge levelled against liturgy—that repetition dulls the consciousness, so that the worshipper becomes an automaton, mentally absent from the worship service.

The phenomenology of habit shows us that this charge is true of the abuse, but not the proper use of liturgical worship. It highlights the perennial temptation, but not the essence of liturgy—indeed, the perennial temptation, but not the essence, of all habit. Habit does not aim to bring about the complete automation of a certain action. Habits do not abolish the intellect; the mind remains active in a healthy habit (Ricœur [1950] 1966, p. 284). Habits provide inclinations but they do not destroy freedom and consciousness. For example, I may have got in the habit of taking a snack in the mid-afternoon, but this does not mean I have no choice about doing so. If I want to lose weight, I can stop, albeit not without effort. Similarly, the repetitive nature of liturgy is never meant to suppress or abolish mental activity. The mind has a crucial place in liturgical worship; consciousness is like the top of a

skyscraper which depends for its very greatness on the habits that support it and hold it in place. As phenomenologist Erazim Kohák writes, "habit is the organ which most increases the efficacy of willing and frees the will from preoccupation with means, enabling it to focus on ends. In providing willing with easy, familiar patterns of action, habit reinforces willing" (Kohák 1966, p. xxii).

Only when habit has become degraded and malfunctional does an action occur "automatically" without any conscious will. "Habit can hold the seed of a threat of falling into automatism," Ricœur warns us (Ricœur [1950] 1966, p. 285). The repetition needed to form a habit can also lead to its becoming degraded into mindlessness: "the temptation to resign my freedom under the inauthentic form of custom, of the 'they', of the 'only natural', of the already seen and already done lies in the very nature of habit" (Ricœur [1950] 1966, pp. 301–2). As Kohák puts it, "the weak or tired will sees in [habit's] easy, readily available pattern not just a tool for effective action but also a relief from responsibility, a substitute for such action" (Kohák 1966, p. xxii). However, this can only happen if the conscious will allows it. The unavoidable weakness in habit is still the result of our free will; that is why Ricœur calls it a temptation. We only fall into the sickness of automation by failing to be attentive to what we are doing, and "failing to be attentive" is still something we have chosen: in other words, we have the freedom to destroy our freedom. "Thus the ultimate significance of habit," summarises Kohák, "depends on the effort which determines whether the will uses [it] or yields to [it]" (Kohák 1966, p. xxii). We may use the gift of habit for our good or our degradation.

The temptation of liturgy is therefore the temptation of habit: to disengage the mind and let the words and actions roll on by themselves without conscious intention. The danger that liturgy will become mindless and meaningless cannot be avoided and remains a permanent weakness. However, this temptation is intrinsic to the structure of liturgy precisely because it is meant to involve the mind, of which free will is an intrinsic aspect. The only way to remove this temptation would be to remove free will, yet that is what the temptation is—the temptation to remove free will and lazily "go with the flow" without consciously engaging. Therefore, those who criticise liturgy because its repetition is mindless are in fact criticising the *corruption* of liturgy, not liturgy itself.

How does the phenomenology of habit answer the charge that liturgy cultivates hypocrisy? To acquire a habit we must perform actions that are not yet natural, but that we hope will become natural. This means there is an in-between stage when the action has not yet become natural. In a popular TED talk social psychologist Amy Cuddy argues that the true motto should not be "fake it till you make it," but "fake it till you *become* it." Her example is that if we adopt a bodily position associated with confidence, we will start to feel more confident, which will make us *be* truly confident in treating other people (Cuddy 2012, emphasis original). C.S. Lewis makes a similar point about the moral life as a whole: "do not waste time bothering whether you 'love' your neighbour; act as if you did. As soon as we do this we find one of the great secrets. When you are behaving as if you loved someone, you will presently come to love them" (Lewis 2009, p. 131). We can easily confuse the practical wisdom of "fake it till you become it" with plain and simple "fakeness" or dishonesty. The difference has to do with our intentions; once again, the conscious will is the pinnacle of liturgy, and habit is there to support it, not to undermine it.

Similarly, to view liturgy as aimed at habit-formation explains why its opponents condemn it as legalism. To form a good habit requires self-discipline, which means keeping to a regime that may feel unnecessarily strict at times. This is illustrated well in a scene from Graham Greene's novel, *The Power and the Glory*, where a Protestant confronts a Catholic with the apparent legalism of Catholicism's many rules: "It seems to me you people make a lot of fuss about inessentials. Fasting . . . fish on Friday . . . " The Priest ponders for a moment and then replies: "After all, Mr. Lehr, you're a German. A great military nation. . . . You understand—discipline is necessary. Drills may be no good in battle, but they form the character" (Greene [1940] 2010, p. 160). In fact, the difference between legalism and self-discipline is not visible to the external observer. The difference lies in the *reason* for

keeping the rules. A legalist keeps the rules for its own sake, seeing rule-keeping as a virtue in itself, and probably judging others who do not keep the rules, whereas in self-discipline they are kept with a purpose that lies beyond the rules: the goal of "forming the character" as Greene puts it. Once again it is intentionality of the will that keeps liturgy from falling into its own corruption.

Finally, to view liturgy as a habit might explain why so many people who are not familiar with liturgy find it boring and hard to engage with. Liturgy is a complex genre with its own patterns, rhythms, and movements that are not obvious to the newcomer. As with any genre, it takes time and practice to become sufficiently "at home" with it to understand what is going on from the inside. To use an example from another genre, people unfamiliar with Hollywood movies can find the rapid succession of scenes bewildering and confusing, because they have not picked up the "cues" that indicate the progression of the story. My own experience confirms that this is also true for liturgy. When I first went to a liturgical service I felt similarly bewildered. It seemed like a random succession of antiphonies, songs, actions, and recitals clumsily bundled together. I only began to "get" it when someone interpreted to me the logic of the liturgical sequence, and even then I had to attend approximately thirty liturgical services before I began to experience the benefit of them. As Brittingham writes:

> Meaningful orders are not always fully understood immediately. One must participate in these meaningful orders with regularity in order for them to become the norm. . . . To expect a practitioner to understand the necessity of the confession of sin within a liturgical service right away is to assume that such an order is less historical than it actually is. Only through continued practice does the liturgical horizon make liturgical practice first intelligible and then meaningful. (Brittingham 2016, p. 156)

In other words, like every culture with a history, liturgy is an acquired taste. Although it is possible to encounter God through an unfamiliar liturgy, the encounter is made richer and more lasting when we have familiarised ourselves with its language.

None of these points is meant to deny the danger that liturgical repetition can descend into unconscious automatism, hypocrisy, legalism, empty and boring routines without life, and all the things liturgy is accused of. Liturgy uses habit as a powerful tool by which we can change our own nature, a "transformation of the living being by its own activity" (Ricœur [1950] 1966, p. 280). However, liturgy is not magic: the practice of liturgy can instil Christlike virtue in us *if and only if* we practise it with that conscious intention and purpose.

## 7. Conclusion: The Decentring of the Self

We have now examined the five most common objections to liturgical worship. Those who love liturgy and even those who take it for granted have important lessons to learn from each of these objections. Yet, from a phenomenological point of view, liturgy still emerges as the form of worship best suited to our finite human condition. In this conclusion I want to draw together the preceding threads to show how they point to the ultimate purpose of liturgy: to displace us from the centre of our lives and to place God there instead.

My position differs from the commonly held view that worship is ultimately "for God" in such a way as to distinguish it from all other elements of life. For example, Byars claims that "worship is an offering to God, 'a sacrifice of praise and thanksgiving' . . . It is simply an end in itself, with no practical function or need for justification" (Byars 2008, p. xvi). He later adds that worship also "has the effect over time of forming and shaping the worshipers in a particular orientation vis-à-vis God" (Byars 2008, p. xvi). However, this still makes the formative aspect of worship seem like an accidental bolt-on, an unnecessary yet useful side-effect. In a sense, I agree that worship is "for God" by default, only because everything in our lives is to be done *ad majorem dei gloriam*. However, that is not what makes worship different from gardening, breastfeeding, or accountancy. God does not need our worship: "if I were hungry, I would not tell you, for the world and all that is in it is mine" (Psalm 50:12 NRSV).

Rather, I suggest that the unique purpose of worship is precisely *to remind us* that everything is "for God" and thus to orient the whole of our lives towards him. Therefore, in one sense, liturgy is "for us" in the sense that its purpose has to do with us. However, in another sense, the very purpose of liturgy is to divest us of our self-centred way of living in which we are only interested in what can be of value "for us." The experience of liturgy is one in which everything, including liturgy, is seen as "for God." "One's centre must not be in oneself but in God," writes Marcel, "outside of that there is no religion" (Marcel [1927] 1952, p. 237). The ultimate goal of liturgy is to displace us from being the centre of our own concerns, and to allow God to take up his rightful place at the centre. This does not mean a total annihilation of the self, but rather a *reorientation* that restores us to *our* rightful place, the only place where we may truly find peace and flourish as who we were created to be.

We see this self-displacement at work in the liturgical calendar, the feasts and fasts that take place regardless of how the individual worshippers happen to feel at that moment. We may have our own personal reasons to be happy or sad, joyful or grieving, and of course God cares deeply about those things. However, in the liturgical calendar we join an ancient worldwide community that is thousands of years old, that commemorates events far bigger and more significant *for each of us* than anything that will happen in our lifetimes. As Gschwandtner puts it, "In liturgy we lose our own individuality and identity, our concern with ourselves, the organization of the entire world around 'me'. We are no longer the ones who open the world, but become dispossessed, swallowed up in the larger story" (Gschwandtner 2019, p. 46). We are caught up in a cycle of events bigger than our individual selves: our own personal reasons for happiness or sadness are not what matters here, and there is something liberating in that. To remember that Jesus is risen, and to celebrate, regardless of what's going on in our life. Additionally, to remember the suffering and evil in the world, regardless of our own personal successes. It helps us escape our own little world and personal worries and concerns, and throws us into the bigger world and all that is going on and has going on throughout.

We see this self-displacement in the call to an authenticity that is not based on feelings but on conformity to the will of God in our lives. The truly authentic person is not the one who is "true to him/herself," because we are not stable, nor could we be even if we were without sin. We are not the source of our own subjectivity. As Ricœur puts it, "the deepening of subjectivity calls for a second Copernican revolution which displaces the center of reference from subjectivity to Transcendence. I am not this center and I can only invoke it and admire it in the ciphers which are its scattered symbols" (Ricœur [1950] 1966, p. 472). The truly authentic person is the one who is true to the highest reality there is, the source of all reality, the transcendent Lord of all the earth. Yet, we cannot see him except through the symbols of created things, symbols that point beyond themselves to a reality that exceeds our visible horizon.

We see this self-displacement in liturgy's invitation to a freedom and a spontaneity that are not limitless, but are bounded by a divine order and structure. It is a strange paradox that to abandon ourselves to our own freedom is to enslave ourselves to passions. Marcel writes that "our freedom has the power to deny itself when it believes it is affirming itself, to become lost in an *impasse* while it claims to be expanding. This is exactly what happens when it deifies itself in fact—without always being fully aware of its act of self-divinization, i.e., when it claims that the world revolves around itself" (Marcel [1940] 1982, p. 30, italics original). Yet, this is what the world tries to entice us to think. Advertising, capitalism, consumerism, our own egos—all conspire to persuade us that we are at the centre, that our needs and desires are what truly matter. They seduce us into prioritising our own comfort and security, offering us illusory promises and mirages and promising that only by indulging ourselves will we be free. These narcotic illusions of worldly life can seem so real while we are in them. We worship to conform our hearts and minds to the divine order of things in which alone our freedom may be truly free, to re-immerse ourselves in the reality of who God is. We worship to reorient our lives toward him, which so easily

and quickly become oriented toward ourselves. We worship to remember that he is the only one who truly satisfies, that we find our rest in him alone (cf. Psalm 62:1).

The point of liturgy is not to leave us without any self at all. Ricœur's metaphor of a Copernican revolution is instructive in this regard. Although earth is not the centre, it still belongs in the solar system with its own gravity and its own satellites. There is a depth and a dignity to being human that liturgy aims to recognize and to restore if it has been damaged. Yet, the ultimate point of liturgy is to take the focus off ourselves and to bring before our eyes the great events in the Christian story of salvation, to remind us of the reason we are here on this earth, to remind us of the vast story God is telling of which we play only the tiniest and least significant role, and to remind us what that role is and why we are most fulfilled, most fully alive, when we participate in that story with all of our hearts.

**Funding:** This research received no external funding.

**Conflicts of Interest:** The author declares no conflict of interest.

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
