# Peer review of "How Can Phenomenology Address Classic Objections to Liturgy?"

_religions, doi:10.3390/rel12040236_

Round 1

Reviewer 1 Report

This paper presents an original research question that is well argued. Also, I appreciate that author constructive-critically applies phenomenology for mediating between competing theological positions. In terms of methodology, the paper is sound. 

With regard to the theme itself, I am no qualified to judge the anti-liturgical movement among Protestant (mostly Evangelical) theologians while this is for the first time that I encounter such a debate. However, the author presents a well-balanced exposition of this debate and his evaluation of the both sides is critical and constructive. 

My only content-related critical remark is the following: The author presents the conclusion that liturgy is an important, even indispensable theological feature for it plays a crucial role in the 'decentering of the subject'; something that is, in author's opinion, a desirable theological goal. Although this position is well-argued and supported with references, for example, to Crina Gschwandtner and Paul Ricoeur, I believe that the concluding position of the author could, and should be a bit more balanced. Let me explain: Phenomenology as the discourse 'from below' is rightly used by the author as the critical counter-pole to the evangelical theology 'from above'. Phenomenology helps to explain the role of our humanity in the process of worshiping God, and thus establishes a proper balance between the human (philosophical) and the divine (theological). However, if we take seriously the decentering tendency of phenomenology (and I agree that there is such a tendency), we need also to take seriously the major reason behind this decentering; that is, the final recollection of our self. In line with the phenomenological practice, we bracket the natural attitude, we decenter our ego and we let the world appear as it is in order to rediscover this world and our position in the world. I believe that the same dynamism should be applied on the liturgy. In other words, I would like to see also this other side of the decentering of the self in liturgy that returns humankind to the self (without projecting a total discrepancy between the human and the divine). The authors such as Jean-Yves Lacoste (and his concept of liturgical existence) or Emmanuel Falque (who has written extensively on the topic) would deserve attention.  

I believe that a bit more balanced conclusion (which would be, in fact, in line with the rest of the paper) would be beneficial. I leave it on the author's decision whether s/he would like to include this criticism into his paper. 

Author Response

This reviewer has suggested only one point for improvement. In response, I have added two discussions in the conclusion that balance the excessive focus on decentring. 

At the end of the 3rd paragraph of the conclusion, I have added the following: "This does not mean a total annihilation of the self, but rather a reorientation that restores us to our rightful place, the only place where we may truly find peace and flourish as who we were created to be."

At the beginning of the final paragraph, I have added the following: "The point of liturgy is not to leave us without any self at all. Ricœur’s metaphor of a Copernican revolution is instructive in this regard. Although earth is not the centre, it still belongs in the solar system with its own gravity and its own satellites. There is a depth and a dignity to being human that liturgy aims to recognize and to restore if it has been damaged. "

Reviewer 2 Report

None.

Author Response

This reviewer has not left any visible comments so I have nothing to respond to. 

Reviewer 3 Report

This piece faces me with a dilemma. I find its position compelling throughout. Each defence of liturgy is one I share, and I wholeheartedly concur with the conclusion. But I find that I do so from a personal, Christian perspective rather than a scholarly one. (I know in some sense these are inseparable, but I hope you will see what I mean). My qualms with the article have to do with its loose use of phenomenology. Most compelling are the references to Merleau-Ponty, Marcel, and Gschwandtner, all thinkers working in the phenomenological tradition. But in a lot of cases, the critique is not necessarily phenomenological (e.g., that the claims for spontaneity against liturgy are self-defeating), and sometimes it is merely semantic or stipulative, e.g., the discussion about authenticity as feeling vs accord between words and actions. 'Authenticity' is a contested term within the phenomenological tradition. See Heidegger, for example, where it's a matter of resoluteness, but one that's supposed to have some feeling behind it too! Another area of concern is the normative vs the descriptive. Phenomenology is chosen here because it tells us "what is" and not "what ought to be." (NB, a case where early modern empiricism fits the same description). But the arguments in favour of liturgy are themselves grounded in the normative claim that our practice *ought* to match with our true, finite nature, which is disclosed by phenomenology (when it's not simply articulated by Paul!) The distinction between our finitude and sinfulness is used a lot as an argumentative clincher, but this needs to be spelled out more. In other words, it's not fortuitous that phenomenology happens to be the right method for the case; there is a normativity built in to the author's 'natural' perspective. "That's just how we are" in this case entails the normative claim that this is how we ought to be. Many cogent points are made in the piece. Yes, if everything is holy, nothing is -- especially since, if I'm not mistaken, the root terms for 'blessed' and 'holy' mean 'set aside'. How can everything be set aside? There must be some distinction, even if we are called to sanctify the world (here again, though, this presupposes work to be done!) But then there are some gross generalisations that seem out of place, e.g., baby boomers make it all about how they feel. It's obviously ok to defend liturgy against objections using a broadly phenomenological frame, but sometimes the piece feels borderline partisan. Main suggestions for improvement: 1) more clarity about how you are using phenomenology; 2) more conceptual work on some of the key claims, e.g., finitude vs sinfulness; 3) be clear that the article is programmatic, not definitive. It seems to me each of the 5 points could be subjected to article-length phenomenological treatment.

Author Response

Point 1: loose use of phenomenology

Response: I have added phenomenological analyses to the two article sections which lacked them, and I have lengthened the phenomenological analysis in a third section. These three additions can be found in lines 112-161, 458-480, and 635-648. 

Point 2: normative vs. descriptive

Response: In lines 212-221, I have elaborated on where I distinguish between "what is" (phenomenology) and "what ought to be" in order to clarify how I transition from one to the other. I do not agree with the reviewer that "there is a normativity built in to the author's 'natural' perspective. 'That's just how we are' in this case entails the normative claim that this is how we ought to be." I have tried to explain in those lines that I do not see phenomenology as offering any normative statement. Its description of the human condition (e.g. that we naturally gravitate towards order) does not answer the question of whether we should accept this feature of ourselves as how we ought to be or fight against it as how we ought not to be. After all, we are all naturally selfish, yet this does not imply that we ought to be. 

Point 3: gross generalisations that seem out of place

The only example given by the reviewer was of a quotation, not my own words. In the context, I was not implicitly agreeing with this quotation - I later critique it although granted I do not critique that particular aspect of it. 

Point 4: be clear that the article is programmatic, not definitive.

Response: I have added a sentence in the introduction (lines 46-50) with a declaration of my own position, saying that the article "does not pretend to be impartial – it is written by someone who is himself a convert to liturgy – but it does make every effort to treat the opposing arguments fairly and do justice to them. Nor does it pretend to be comprehensive – much more could be said on each of these points. My goal is to provide a sketch of each objection and a brief insight into how phenomenology reframes it."

Round 2

Reviewer 3 Report

Much improved! The appeal to phenomenology is better motivated now, and the expanded material on Marcel and Ricoeur give it much more depth. I also think the frankness up front about the motivations makes the piece stronger.